# Structural basis of human ghrelin receptor signaling by ghrelin and the synthetic agonist ibutamoren

Heng Liu[1], Dapeng Sun [1], Alexander Myasnikov [2], Marjorie Damian[3], Jean-Louis Baneres [3], Ji Sun [2✉] & Cheng Zhang [1✉]

The hunger hormone ghrelin activates the ghrelin receptor GHSR to stimulate food intake and growth hormone secretion and regulate reward signaling. Acylation of ghrelin at Ser3 is required for its agonistic action on GHSR. Synthetic agonists of GHSR are under clinical evaluation for disorders related to appetite and growth hormone dysregulation. Here, we report high-resolution cryo-EM structures of the GHSR-$G_i$ signaling complex with ghrelin and the non-peptide agonist ibutamoren as an investigational new drug. Our structures together with mutagenesis data reveal the molecular basis for the binding of ghrelin and ibutamoren. Structural comparison suggests a salt bridge and an aromatic cluster near the agonist-binding pocket as important structural motifs in receptor activation. Notable structural variations of the $G_i$ and GHSR coupling are observed in our cryo-EM analysis. Our results provide a framework for understanding GHSR signaling and developing new GHSR agonist drugs.

[1] Department of Pharmacology and Chemical Biology, School of Medicine, University of Pittsburgh, Pittsburgh, PA 15261, USA. [2] Department of Structural Biology, St. Jude Children's Research Hospital, Memphis, TN 38120, USA. [3] Institut des Biomolécules Max Mousseron, CNRS, Université de Montpellier, ENSCM, Montpellier, France. ✉email: Ji.Sun@stjude.org; chengzh@pitt.edu

The human ghrelin system is a critical component of the gut-brain axis to regulate energy homeostasis and reward signaling[1–3]. Ghrelin is a 28-amino acid orexigenic peptide hormone generated in the gut commonly regarded as the hunger hormone or survival hormone[1–4]. It mainly signals through the growth hormone secretagogue receptor (GHSR, or ghrelin receptor), a class A G protein-coupled receptor (GPCR). Ghrelin signaling through GHSR stimulates growth hormone secretion and food intake under nutritional or physiological challenges[1,2]. In addition, GHSR plays critical role in the modulation of stress and anxiety[5,6] and the regulation of dopamine signaling and thus reward pathways in the CNS[7,8]. Therefore, GHSR is a highly pursued drug target[9]. A GHSR inverse agonist named PF-5190457 is under clinical investigation to treat alcohol use disorder[10,11]. On the other hand, pharmacological stimulation of GHSR by agonists represents an emerging and exciting avenue to address disorders related to appetite, gastric emptying, and growth hormone dys-regulation. Relamorelin and anamorelin are two synthetic peptide agonists of GHSR that are in different stages of clinical trials to treat diabetic gastroparesis and cancer-related anorexia-cachexia[12–16]. Ibutamoren, also known as MK-0677 or LUM-201, is a synthetic small-molecule GHSR agonist used as a growth hormone secretagogue in many disease settings[9,17–20].

The ghrelin-GHSR signaling system exhibits several unique properties compared to other GPCR signaling systems. Acylation of ghrelin, usually octanoylation at position-3 serine, is required for its action on GHSR[3]. Such a modification is achieved by the ghrelin O-acyl-transferase (GOAT) in vivo[21–23]. Acylated ghrelin can activate GHSR to signal through multiple protein partners, including the $G_{q/11}$ and $G_{i/o}$ families of G proteins and β-arrestins[24]. Several studies showed that different GHSR signaling pathways regulate distinct physiological functions[25–27]. The GHSR and $G_i$ signaling pathway has been linked to attenuated glucose-induced insulin release by ghrelin[25], while the appetite stimulation is mainly mediated by $G_{q/11}$ signaling[26]. GHSR also exhibits a remarkably high constitutive activity, which is important for physiological growth hormone regulation[28]. Mutations of the GHSR gene that result in loss of GHSR constitutive activity have been associated with the familial short stature syndrome[29]. In addition, the constitutive activity of GHSR is negatively regulated by several endogenous mechanisms. Liver expressed antimicrobial peptide 2 (LEAP2) has been characterized as a GHSR inverse agonist or antagonist[30]. Melanocortin receptor accessory protein 2 (MRAP2), a single-pass transmembrane protein, can directly associate with GHSR to modulate its activity[3,31].

Due to the physiological and pathological significance of the ghrelin-GHSR system, intensive research effort has been devoted to the molecular understanding of ghrelin action and GHSR signaling. Biophysical studies and structural modeling were performed to investigate ghrelin recognition by GHSR[32,33]. Recently, a crystal structure of highly engineered GHSR at an inactive conformation bound to an antagonist, Compound 12 (C12), has been reported to probe a possible ghrelin recognition mechanism[34]. Yet, the molecular mechanism underlying GHSR signaling by diverse peptide and non-peptide agonists still remains elusive. Here, we report two cryo-electron microscopy (cryo-EM) structures of human GHSR in complex with $G_i$ and two agonists, the endogenous ligand ghrelin and the non-peptide synthetic agonist ibutamoren. Structural analysis together with mutagenesis data revealed mechanisms underlying agonism of different types of GHSR agonists and GHSR-$G_i$ coupling.

## Results and discussion

**Structure determination of two GHSR-$G_i$ complexes and overall structures.** We used wild-type human GHSR and human

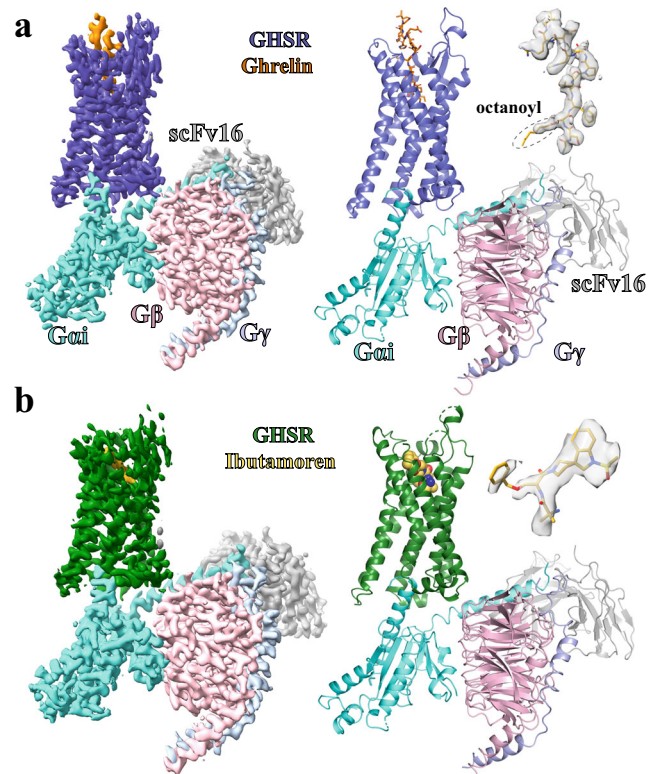

**Fig. 1 Overall structures of the GHSR-$G_i$ complexes. a, b** Cryo-EM density maps of the complex with ghrelin (orange) and ibutamoren (yellow) and overall structures. GHSR bound to ghrelin and ibutamoren is colored in blue and green, respectively. $G_{αi}$, $G_β$, and $G_γ$ subunits are colored in cyan, pink, and light blue, respectively. ScFv16 is colored in gray. The density maps of the two agonists are shown in light gray. The octanoyl group in ghrelin is circled.

$G_{αi}$, $G_{β1}$, and $G_{γ2}$ to assemble the complexes with two agonists. No modification was introduced to the $G_i$ heterotrimer except for the amino-terminal 6xHis tag in $G_{β1}$. We used apyrase to hydrolyze GDP in order to form stable nucleotide-free complexes. We also added an antibody fragment, scFv16, to further stabilize the $G_i$ heterotrimer[35]. The structures of the GHSR-$G_i$-scFv16 complexes with ghrelin and ibutamoren were both determined to an overall resolution of 2.7 Å (Fig. 1a, b, Supplementary Fig. 1, and Supplementary Table 1). The cryo-EM maps allowed modeling of most regions of GHSR and $G_i$ heterotrimer, ghrelin residues Gly1-Val12 (residues in ghrelin are referred to by three-letter names, and residues in GHSR and other GPCRs are referred to by one-letter names hereafter) together with the octanoyl group, and the entire ibutamoren molecule (Fig. 1a, b and Supplementary Fig. 1). For ibutamoren, the overall conformation depicted in our structure is strongly supported by the density map. However, due to the limitation of resolution of the density map, subtle structural variations of three arms of ibutamoren are possible.

We also observed cryo-EM densities likely corresponding to cholesterol molecules bound to GHSR in both structures (Supplementary Fig. 2a). Further experiments showed that cholesterol could positively regulate the binding of ghrelin to GHSR and the activity of GHSR in reconstituted lipid nanodiscs (Supplementary Fig. 2b, c), suggesting a potentially significant physiological role of cholesterol in ghrelin signaling.

**Ghrelin and ibutamoren recognition.** In our structure, the N-terminal part of ghrelin adopts an extended conformation and

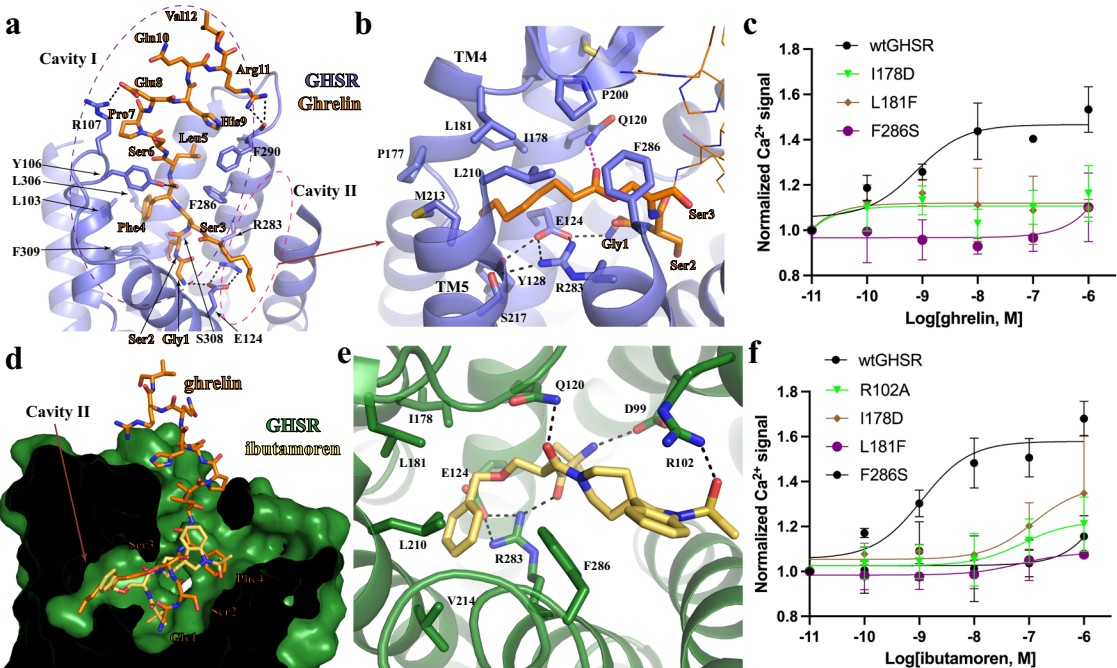

**Fig. 2 Binding pockets for ghrelin and ibutamoren. a** Ghrelin binding in the cavity I and II. Ghrelin residues are labeled in orange. GHSR is colored in slate. **b** Binding of the octanoyl group of ghrelin in the cavity II. **c** Dose-dependent action of ghrelin on the wild type GHSR (wtGHSR) and mutants. **d** Alignment of ghrelin and ibutamoren. GHSR bound to ibutamoren is colored green. Ibutamoren is shown as yellow sticks. **e** Ibutamoren binding pocket. **f** Dose-dependent action of ibutamoren on wtGHSR and mutants. In **c** and **f**, agonist-induced GHSR signaling was measured by $Ca^{2+}$ mobilization assay. Each data point represents mean ± S.D. from three independent assays. Polar interactions are shown as dashed lines in (**a**, **b**, and **e**).

inserts into a deep pocket of GHSR (Fig. 2a); such a binding mode is different from the predicted ghrelin binding mode based on the previous crystal structure[34]. It was suggested that the salt bridge between E124[3.33] and R283[6.55] (superscripts represent Ballesteros−Weinstein numbering[36]) of GHSR divides the ghrelin-binding pocket into two cavities, cavity I and II, and the octanoyl moiety of ghrelin was predicted to occupy a hydrophobic crevasse between TM6 and TM7 in the large cavity I[34]. In our structures, we also observed the salt bridge between E124[3.33] and R283[6.55]. However, different from the predicted binding mode of ghrelin, all the peptide moiety of ghrelin occupies the large cavity I while the octanoyl moiety attached to Ser3 of ghrelin extends into the small cavity II in the structure of ghrelin-bound GHSR (Fig. 2a). Furthermore, in both of our structures, TM7 shifts towards TM6 compared to that in the inactive GHSR, closing the previously observed crevasse in the inactive structure (see "Discussion" below).

Ghrelin engages in extensive polar and hydrophobic interactions with GHSR in the cavity I. From the bottom region to the extracellular side, the side chains of ghrelin residues Gly1 and Glu8 form salt bridges with the side chains of E124[3.33] and R107[ECL1] of GHSR, respectively; the side chains of Ser2 and Arg11 and the main chain carbonyl of Phe4 of ghrelin are hydrogen-bonded by the main chain carbonyl groups of F290[ECL2] and S308[7.38] and the side chain of Y106 of GHSR, respectively (Fig. 2a). In addition to these polar interactions, in the middle region of the cavity II, Phe4 of ghrelin packs against GHSR residues L103[2.64], L306[7.36], and F309[7.39] to form hydrophobic interactions (Fig. 2a). Leu5 of ghrelin also forms hydrophobic interactions with GHSR residues F286[6.58] and F290[6.62] (Fig. 2a).

The octanoyl moiety of ghrelin mainly occupies the cavity II with an extended conformation pointing towards the cleft between TM4 and TM5 (Fig. 2b). We observed a strong density corresponding to the proximal half of the octanoyl moiety, while the density for the distal half was not well resolved, suggesting a

flexible nature of this part (Fig. 1a). Of note, the cavity II exhibits an amphipathic environment (Fig. 2b). As a result, the $C_8$ fatty acid chain of the octanoyl moiety sits on top of an extensive polar interaction network at the bottom of the cavity II including E124[3.33] and R283[6.55] while being capped by hydrophobic GHSR residues from the upper region of the cavity II (Fig. 2b). The acyl group of the octanoyl moiety sticks towards and likely interacts with the side chain of Q120[3.29] of GHSR. Previous mutagenesis studies showed that individual mutations of E124[3.33], R283[6.55], and Q120[3.29] to non-polar residues led to decreased potency of ghrelin[34,37]. We also showed that mutations of hydrophobic residues in the cavity II to larger or polar residues, which potentially disrupt the binding of the octanoyl moiety, resulted in compromised GHSR signaling induced by ghrelin (Fig. 2c and Supplementary Fig. 3a). In addition, we measured the binding of ghrelin to wild-type GHSR and mutants. Individual mutations of GHSR residues I178[4.60], L181[4.63], and F286[6.58] that interact with the octanoyl moiety of ghrelin to alanine led to the decreased affinity of GHSR for ghrelin (Supplementary Fig. 3b). Mutations of the salt bridge residues E124[3.33] and R283[6.55] to either alanine or glutamine could almost abolish ghrelin binding (Supplementary Fig. 3b). Collectively, those mutagenesis data support the binding mode of ghrelin observed in our structure.

The non-peptide agonist ibutamoren also samples both cavity I and cavity II (Fig. 2d). It mainly occupies the bottom space of the ghrelin-binding pocket similar to the antagonist C12. The structures of ghrelin- and ibutamoren-bound GHSR share similar overall conformation with a root-mean-square deviation (RMSD) of the Cα atoms at 1.23 Å. Subtle structural differences can be observed at the upper region of the binding pocket including ECL2 and 3 (Supplementary Fig. 3c), likely caused by the binding of Leu5-Val12 of ghrelin in this region. The three arms of ibutamoren point towards three different directions, mimicking the first four residues of ghrelin, Gly1-Phe4, including the octanoyl moiety (Fig. 2d). The phenyl group of ibutamoren sits in

the cavity II and forms hydrophobic interactions with surrounding hydrophobic residues I178[4.60], L181[4.63], and L210[5.36] of GHSR similarly to the octanoyl moiety of ghrelin (Fig. 2e). The other two arms of ibutamoren stick towards the bottom region of cavity I and the extracellular surface, respectively (Fig. 2e and Supplementary Fig. 3d). Multiple hydrogen bonds are observed at the bottom of the binding pocket between ibutamoren and GHSR residues D99[2.60], Q120[3.29], and R283[6.55] (Fig. 2e). Of note, the bottom region of cavity I exhibits a negatively charged environment to accommodate the smallest arm of ibutamoren with two amine groups (Supplementary Fig. 3d). The largest arm of ibutamoren containing an indoline-3,4′-piperidine ring structure forms hydrophobic and cation-π interactions with the side chains of GHSR residues F286[6.58] and R102[2.63], respectively. R102[2.63] also forms a hydrogen bond with a hydroxyl group attached to the indoline ring of ibutamoren. Mutations of hydrophobic residues in the cavity II caused similar functional consequences to both ibutamoren and ghrelin (Fig. 2f). In addition, mutation of R102[2.63], which forms hydrogen bonding and cation-π interactions with ibutamoren, resulted in compromised action of ibutamoren as well (Fig. 2f).

**Distinctive activation mechanism of GHSR.** GHSR adopts an active conformation in our structures, which is stabilized by the G$_i$ protein. Both extracellular and cytoplasmic regions of active GHSR showed significant structural differences compared to the inactive GHSR in the crystal structure reported before[34]. At the extracellular region, the most prominent difference observed lies in TM7 (Fig. 3a), which is unusual for class A GPCRs[38]. Compared to that in the inactive GHSR, the extracellular segment of TM7 in both active GHSR structures with ghrelin and ibutamoren extends by two more helical turns and moves towards TM6 (Fig. 3a and Supplementary Fig. 4a). However, such a large conformational change is not likely to be caused directly by agonist-binding since there is little interaction between TM7 and both agonists. We further mutated individual residues at the extracellular end of TM7 from E297 to I300 to proline and measured ghrelin-induced GHSR activation. Our results suggested that this part of TM7 is not critical for the agonistic action of ghrelin (Supplementary Fig. 4b), consistent with little involvement of this region in ghrelin binding based on our structure. It is possible that the disordered extracellular region of TM7 observed in the crystal structure of inactive GHSR was caused by crystal packing[34].

We also observed a notable shift of the extracellular half of TM6 in the active GHSR in comparison with the inactive GHSR (Fig. 3a). In the core region of GHSR, we observed remarkable conformational changes of highly conserved residues V131[3.40], P224[5.50], F272[6.44], and W276[6.48] (Fig. 3b). Conformational

changes of these conserved residues especially the 'transmission switch' residues F272[6.44] and W276[6.48] have been suggested to link the extracellular agonist-binding to the cytoplasmic receptor activation and G protein-coupling for class A GPCRs (Supplementary Fig. 4c)[38–40]. The cytoplasmic region of GHSR in our structures exhibits typical features of activated GPCRs including a large outward displacement of TM6 and an inward movement of TM7 (Fig. 3c)[38]. In addition, the cytoplasmic regions of TMs 1-4 together with ICL1 and ICL2 all showed notable movements. Surprisingly, the cytoplasmic region of TM5 in the inactive and active structures are almost identical (Fig. 3c). This is in contrast to most of other class A GPCRs, for which TM5 usually undergoes notable conformational changes at the cytoplasmic region during receptor activation[38]. Several residues in the middle region of TM5 from F221[5.47] to P224[5.50] do exhibit large conformational changes in the active GHSR compared to those in the inactive GHSR (Supplementary Fig. 4d). However, these changes do not translate to a significant displacement of the cytoplasmic region of TM5.

Our structural analysis suggested a receptor activation mechanism involving conformational changes of the salt bridge pair E124[3.33] and R283[6.55] and a unique structural motif underneath. Structural alignment of the ligand-binding pockets for the two agonists and the antagonist C12 suggests that agonist-binding causes a subtle change in the salt bridge between E124[3.33] and R283[6.55] (Fig. 4a). Specifically, the side chain of R283[6.55] moves towards F279[6.51] and H280[6.52] due to steric effects with the octanoyl chain of ghrelin or the phenyl group of ibutamoren (Fig. 4a, b). This conformational change further transduces through F279[6.51] and H280[6.52] to cause rearrangement of an aromatic cluster formed by residues W276[6.48], F279[6.51], H280[6.52], and F312[7.42] (Fig. 4b). As a result, W276[6.48] toggles to further induce significant displacement of F272[6.44] and the cytoplasmic segment of TM6, resulting in the activation of GHSR (Fig. 4b and Supplementary Fig. 4c). Such a notion of receptor activation was supported by previous mutagenesis studies, which showed that individual mutations of residues F279[6.51], F312[7.42], and W276[6.48] in the aromatic cluster significantly lowered the potency of ghrelin in activating GHSR[34,37]. Our proposed receptor activation mechanism also suggests that the conformation of a GHSR ligand around the E124[3.33]-R283[6.55] salt bridge in the cavity II is an important structural determinant for ligand efficacy, providing a molecular foundation for designing novel ghrelin agonists.

**Molecular details of G$_i$-coupling to GHSR.** The G$_i$-coupling is almost identical in both structures of the GHSR-G$_i$ complexes with two agonists. The major interaction site with GHSR is at the C-terminal half of α5 of Gα$_i$ (Fig. 5a, b). Hydrophobic residues I344, L348, L353 of Gα$_i$ line on one side of α5 to form

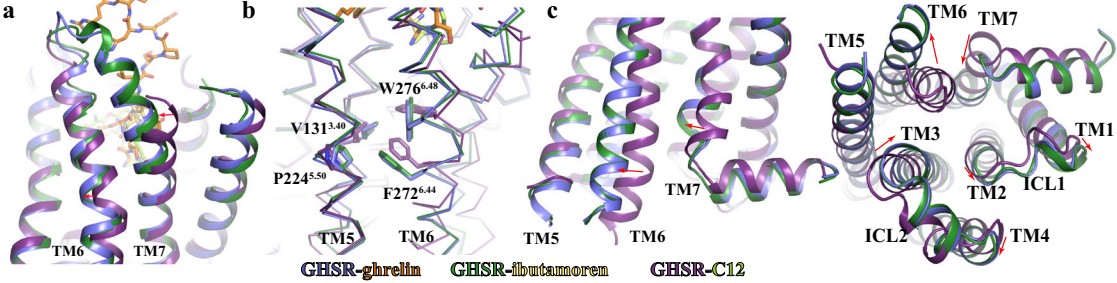

**Fig. 3 Active conformation of GHSR.** Conformational changes of the extracellular regions of TM6 and TM7 (**a**), the highly conserved transmission switch residues (**b**), and cytoplasmic regions of TMs (**c**) in the active GHSR relative to them in the inactive GHSR. Two active GHSR bound to ghrelin and ibutamoren and the inactive GHSR bound to C12 (PDB code 6KO5) are colored in blue, green, and purple, respectively. Ghrelin, ibutamoren and C12 are shown as orange, yellow and lemon sticks, respectively. Red arrows indicate changes of residues or regions from inactive to active states.

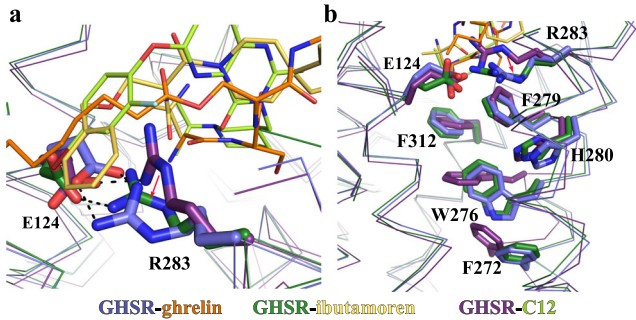

**Fig. 4 Conformational changes of critical motifs near the agonist-binding pocket. a** Different conformations of the E124-R283 salt bridge in the active structures of GHSR with ghrelin and ibutamoren and the inactive structure of GHSR with C12. The salt bridge interactions are shown as dashed lines. **b** Rearrangement of the aromatic cluster residues W276, F279, H280, and F312 in the active and inactive GHSR. Two active GHSR and the inactive GHSR structures are colored in blue, green, and purple, respectively. Ghrelin, ibutamoren, and C12 are shown as orange, yellow and lemon sticks, respectively.

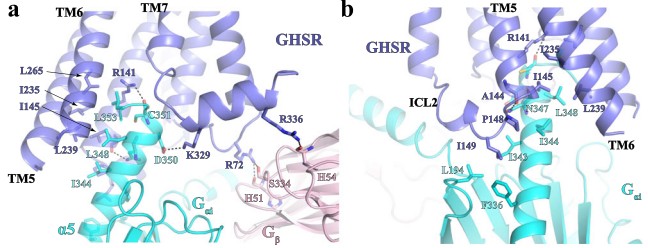

**Fig. 5 GHSR and $G_i$ binding interface. a**, **b** Detailed interactions between GHSR and $G_i$ viewed from two angles. GHSR, $G_{\alpha i}$, and $G_\beta$ are colored in blue, cyan, and pink, respectively. Polar interactions are shown as dashed lines.

hydrophobic interactions with GHSR residues I145[3.54], I235[5.61], I239[5.65], and L265[6.37] (Fig. 5a). The side chains of N347 and D350 of $G_{\alpha i}$ engage in polar interactions with the main chain carbonyl of A144 and the side chain of K329 of GHSR, respectively (Fig. 5a, b). The side chain of R141[3.50] in the conserved DR[3.50]Y motif of GHSR also forms a hydrogen bond with the main chain carbonyl of C351 of $G_{\alpha i}$. In addition, F336, I343, and I344 in α5 and L194 in β2-β3 loop of $G_{\alpha i}$ pack against GHSR residues P148 and I149 in ICL2 to form another set of hydrophobic interactions at the receptor and $G_{\alpha i}$ interface (Fig. 5b). Besides $G_{\alpha i}$, $G_\beta$ also directly interacts with GHSR through multiple polar interactions (Fig. 5a), similar to those observed in the structure of the formylpeptide 2 (FPR2)-$G_i$ complex[41].

Despite distinctive ligand recognition and receptor activation mechanisms, the $G_i$-coupling mode of GHSR is highly similar to those of other closely related neuropeptide GPCRs such as neurotensin receptors (NTSRs)[42]. Previous structural studies on the NTSR1-$G_i$ complex revealed two conformational states of the complex, the canonical and non-canonical states[43]. The structural comparison indicates that the GHSR-$G_i$ complex in our structures resembles the canonical NTSR1-$G_i$ state with a similar $G_i$ interaction profile (Fig. 6a). It is to be noted that our cryo-EM analysis also revealed minor structural classes of particles with different conformations besides the ghrelin- and ibutamoren-GHSR-$G_i$ complexes with the 2.7 Å cryo-EM maps (Supplementary Fig. 1c). We assigned the conformation of the GHSR-$G_i$ complex in the 2.7 Å structures as Conformer 1. 3D reconstruction using particles from one minor class of the GHSR-$G_i$ complex with ghrelin or ibutamoren yielded a 3.5 Å map of a

second conformation, which we assigned as Conformer 2 (Supplementary Figs. 1c, d and 5). We modeled the structure of the ghrelin-GHSR-$G_i$ complex at Conformer 2 (Fig. 6b). Structural comparison with Conformer 1 showed that there is a small but remarkable shift of the entire $G_i$ heterotrimer relative to the receptor with a ~15° rotation of the N-terminal α-helix (αN) and ~10° rotation of the C-terminal α-helix (α5) of $G_{\alpha i}$ in Conformer 2 (Fig. 6b). Despite such differences in the orientation of $G_i$, the structure of ghrelin-GHSR and the interactions between $G_i$ and GHSR stay largely unchanged in Conformer 2. Similar structural variations were also observed for the NTSR1-$G_i$ complex at both canonical and non-canonical states[43]. The different $G_i$-binding modes may imply a highly dynamic nature of $G_i$-coupling to GHSR. However, we cannot rule out the possibility that the structural variations we observed were caused by the non-native buffer conditions in our cryo-EM studies.

In summary, we report two cryo-EM structures of the GHSR-$G_i$ complexes with ghrelin and ibutamoren. Our structures clearly reveal the ligand-binding pocket for ghrelin, where the peptide moiety of ghrelin mainly occupies cavity I while the octanoyl moiety adopts an extended conformation to occupy cavity II. Ibutamoren mimics the first four N-terminal residues of ghrelin including the octanoyl moiety to bind at the bottom region of the ligand-binding pocket. Both agonists cause conformational changes of the salt bridge pair E124[3.33] and R283[6.55] and the aromatic cluster W276[6.48], F279[6.51], H280[6.52], and F312[7.42], which may, in turn, lead to GHSR activation. The overall conformation of the GHSR-$G_i$ complex in our structures highly resembles the canonical state of the NTSR1-$G_i$ complex. The variations of $G_i$-coupling observed for both GHSR and NTSR1 shed light on a potentially highly dynamic nature of $G_i$-coupling to GPCRs.

## Methods
**GHSR expression and purification.** All primers used in this study are listed in Supplementary Table 2. The coding sequence of wild-type human ghrelin receptor (GHSR) was cloned into pFastbac (ThermoFisher) with an N terminal Flag tag followed by a peptide sequence corresponding to the N-terminal fragment of the human β2-adrenergic receptor to increase its expression. The protein was expressed in Sf9 insect cells (Invitrogen) using the Bac-to-Bac baculovirus expression system (ThermoFisher). 1 L cells were lysed by stirring in buffer containing 20 mM Tris-HCl, pH7.5, 0.2 μg/ml leupeptin, 100 μg/ml benzamidine, and 1 μM PF-05190457 (GHSR antagonist) (Tocris). Cell membranes were isolated by centrifugation at 25,000 × g for 40 min. Then the pellet was solubilized in buffer containing 20 mM HEPES, pH 7.5, 750 mM NaCl, 1% (w/v) n-dodecyl-b-D-maltoside (DDM, Anatrace), 0.2% (w/v) sodium cholate (Sigma), 0.1% (w/v) cholesterol hemisuccinate (CHS, Anatrace), 20% (v/v) glycerol, 0.2 μg/ml leupeptin, 100 μg/ml benzamidine, 500 unit Salt Active Nuclease (Arcticzymes) and 1 μM PF-05190457 at 4 °C for 3 h. Insoluble material was separated by centrifugation at 25,000 × g for 40 min. The supernatant was incubated with nickel Sepharose resin (GE healthcare) plus 15 mM imidazole at 4 °C overnight. The resin was washed with 5 column volume of buffer containing 20 mM HEPES, pH 7.5, 500 mM NaCl, 0.1% (w/v) DDM, 0.02% (w/v) CHS, 25 mM imidazole and 1 μM PF-05190457. The protein was eluted with buffer containing 20 mM HEPES, pH 7.5, 500 mM NaCl, 0.1% (w/v) DDM, 0.02% (w/v) CHS, 400 mM imidazole and 1 μM PF-05190457 and loaded onto an anti-Flag M1 antibody resin after adding 2 mM CaCl2. To make the resin, purified M1 antibody was immobilized onto a CNBr-activated Sepharose resin (Cytiva Life Sciences) through coupling of free -NH2 groups to the matrix. The detergent was exchanged to 0.01% (w/v) lauryl maltose neopentyl glycol (MNG, Anatrace) on the M1 antibody resin. To do so, the column bound with GHSR was washed with the following three buffers in a sequential way—buffer 1: 20 mM HEPES, pH 7.5, 100 mM NaCl, 0.05% DDM, 0.02% CHS, 0.05% MNG; buffer 2: 20 mM HEPES, pH 7.5, 100 mM NaCl, 0.02% DDM, 0.02% CHS, 0.07% MNG; buffer 3: 20 mM HEPES, pH 7.5, 100 mM NaCl, 0.01% DDM, 0.02% CHS, 0.1% MNG. In all three buffers, 1 μM ghrelin (Tocris) or ibutamoren (MK-0677, Tocris) was included. Each washing step took 30 min. The protein was finally eluted with buffer containing 20 mM HEPES, pH 7.5, 100 mM NaCl, 0.01% (w/v) MNG, 0.001% (w/v) CHS, 200 mg/ml Flag peptide, 5 mM EDTA and 1 μM ghrelin or ibutamoren, and further purified by size-exclusion chromatography on a Superdex 200 Increase column (Cytiva) using buffer containing 20 mM HEPES pH 7.5, 100 mM NaCl, 0.01% (w/v) MNG, 0.001% (w/v) CHS, and 1 μM ghrelin or ibutamoren. The receptor was collected and concentrated to 5 mg/ml using 100 kDa molecular weight cut-off concentrators (Millipore) for complex assembly.

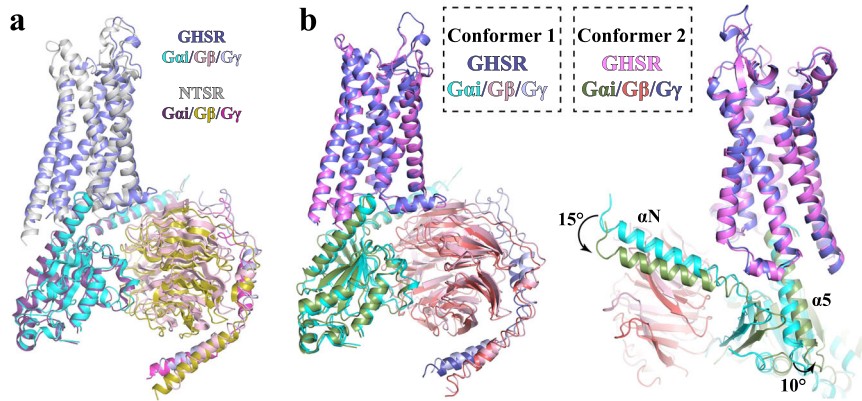

**Fig. 6 Structures of the ghrelin-GHSR-G$_i$ complex in Conformer 1 and 2. a** Structural alignment of the ghrelin-GHSR-G$_i$ complex in Conformer 1 with the neurotensin-NTSR-G$_i$ complex in the canonical state (PDB ID 6OS9). GHSR and NTSR are colored in slate and light gray, respectively. G$_{\alpha i}$, G$_\beta$, and G$_\gamma$ subunits are colored in cyan, pink, and light blue, respectively, in the structure with GHSR, and in dark purple, olive, and warm pink, respectively, in the structure with NTSR. The G$_i$-coupling mode is highly similar. **b** Structural alignment of the ghrelin-GHSR-G$_i$ complex in Conformer 1 and 2. G$_{\alpha i}$, G$_\beta$, and G$_\gamma$ subunits are colored in cyan, pink and light blue, respectively, in Conformer 1, and in forest, ruby, and deep blue, respectively, in Conformer 2. GHSR is colored in slate in Conformer 1 and violet in Conformer 2.

**Expression and purification of G$_i$ heterotrimer.** The wild-type G$_{\alpha i1}$ was cloned in pFastBac vector without any tag, and the virus was prepared using Bac-to-Bac system (Invitrogen). N-terminal 6 × His-tagged human G$_{\beta 1}$, and human G$_{\gamma 2}$ were cloned into pVL1392 vector, and the virus was prepared using the BestBac system (Expression Systems). The heterotrimeric Gi complex was expressed in Sf9 insect cells by co-expressing all three subunits. Cells at a density of $4 \times 10^6$/ml were infected with both G$_{\alpha i}$ and G$_{\beta\gamma}$ virus at ratios of 20 and 1 ml per liter, respectively, at 27 °C for 48 h before harvesting. Cells were lysed in lysis buffer containing 10 mM Tris, pH 7.5, 100 μM MgCl$_2$, 5 mM β-mercaptoethanol (β-ME), 10 μM GDP, 0.2 μg/ml leupeptin, and 150 μg/ml benzamidine. The cell membrane was collected by centrifugation at $25{,}000 \times g$ for 30 min at 4 °C. Cell membranes were solubilized in solubilization buffer containing 20 mM HEPES pH 7.5, 100 mM NaCl, 1% sodium cholate, 0.05% DDM, 5 mM MgCl$_2$, 2 μL CIP, 5 mM β-ME, 10 μM GDP, 10% glycerol, 0.2 μg/ml leupeptin and 150 μg/ml benzamidine. The supernatant was separated by centrifugation at $25{,}000 \times g$ for 30 min and incubated with nickel resin in batch for 1 h at 4 °C. The resin was then washed in batch with solubilization buffer and transferred to a gravity column. The buffer was exchanged on column from solubilization buffer to wash buffer comprised of 20 mM HEPES pH 7.5, 50 mM NaCl, 0.1% DDM, 1 mM MgCl$_2$, 5 mM β-ME, 10 μM GDP, 0.2 μg/ml leupeptin, and 150 μg/ml benzamidine. The protein was eluted in wash buffer with 250 mM imidazole and treated with Lamda Phosphatase (New England BioLab) and Alkaline Phosphatase (New England BioLab) overnight at 4 °C. The protein was further purified with anion exchange chromatography. The low salt buffer was comprised of 20 mM HEPES pH 7.5, 40 mM NaCl, 0.1% DDM, 1 mM MgCl$_2$, 100 μM TCEP, 10 μM GDP. The high salt buffer was prepared by adding 1 M NaCl to the low salt buffer. The pure G$_i$ with an appropriate stoichiometry of three subunits was supplemented by 10% glycerol, concentrated to ~20 mg/ml, flash-frozen in liquid nitrogen, and stored at −80 °C.

**Assembly of GHSR-G$_i$ complexes.** Purified GHSR was mixed with G$_i$ heterotrimer at a 1:1.2 molar ratio. The coupling reaction was initiated by incubation at 25 °C for 1 h and was followed by the addition of Apyrase (New England BioLab) to catalyze the hydrolysis of GDP overnight at 4 °C. This was to form the stable nucleotide-free complex. To remove the excess G$_i$ protein, the mixture was further purified with anti-Flag M1 antibody resin. The complex was eluted using the buffer comprised of 20 mM HEPES pH 7.5, 100 mM NaCl, 0.01% MNG, 0.001% CHS, 1 μM ghrelin or ibutamoren, 200 μg/ml Flag peptide. Finally, a 1.2 molar excess of scFV16 was added to the elution. The GHSR–Gi–scFV16 complex was purified and buffer-exchanged by size exclusion chromatography with buffer containing 20 mM HEPES pH 7.5, 100 mM NaCl, 0.002% MNG, 0.001% CHS, and 1 μM ghrelin or ibutamoren. Peak fractions were concentrated to ~7 mg/ml for cryo-EM data collection.

**Cryo-EM data collection, processing, model building, and refinement.** Cryo-EM grids were prepared with a Vitrobot Mark IV (FEI). Quantifoil R1.2/1.3 holey carbon gold grids (SPI) were glow-discharged for 30 s. Then 3.5 μL of 7 mg/ml protein sample was pipetted onto the grids, which were blotted for 3 s under blot force −3 at 100% humidity and frozen in liquid nitrogen cooled liquid ethane. The grids were loaded onto a 300 keV Titan Krios (FEI) with a K3 direct electron detector (Gatan) and an energy filter. Images of the ghrelin-GHSR-G$_i$ complex dataset were recorded with SerialEM[44] in super-resolution mode with a pixel size of 0.649 Å and a defocus range of −0.8 to −1.6 μm. Movies were recorded during a 2.1 s exposure with 30 ms subframes

(70 total frames) at a dose rate of 1.176 e/Å$^2$/frame. The dataset with the ibutamoren-GHSR-G$_i$ complex was collected with a pixel size of 0.826 Å and a defocus range of −0.8 to −1.6 μm. Movies were recorded during a 3 s exposure with 50 ms subframes (60 total frames) at a dose rate of 1.35 e/Å$^2$/frame.

Both datasets were processed in cryoSPARC[45] using a similar strategy (Supplementary Fig. 1c, d). Briefly, super-resolution image stacks were gain-normalized, binned by 2 with Fourier cropping, and corrected for beam-induced motion using MotionCor2[46]. Contrast transfer function (CTF) parameters were estimated from motion-corrected images using GCTF[47]. Following multiple rounds of 2D classification, ab initio reconstruction was performed in cryoSPARC asking for four classes, which resulted in one good class and three trash classes. Then multiple rounds of heterogenous refinement were performed against the four ab initio models to remove bad particles. Following the CTF refinement and non-uniform (NU) refinement, one last round of heterogenous refinement was carried out using three identical initial models (output of NU refinement). The heterogeneous refinement yielded two slightly different conformations, which were further refined by NU refinement. Resolutions were estimated using the 'gold standard' criterion (FSC = 0.143), and the local resolution was calculated in cryoSPARC.

The coordinates of NTSR, G$_i$, and scFv16 from the structure of the neurotensin-NTSR-G$_i$-scFv16 complex (PDB ID 6OS9) were used as initial models to dock into the cryo-EM maps using Chimera[48]. The structures of GHSR-G$_i$ with both agonists were then built by iterative manual building and adjustment in Coot[49] and real-space refinement in Phenix[50]. The final models were validated by Molprobity[51]. The local resolution maps were calculated by blocres (Supplementary Fig. 1e, f)[52]. To validate our refinement protocol, we used phenix.mtriage[50] to calculate FSC curves of our models against the full map and half maps[53] (Supplementary Fig. 1g, h). Subtle differences between FSC curves calculated based on the two half maps and the full map indicated that our structural models were not over-refined. Data collection, processing, and structure refinement statistics are listed in Supplementary Table 1.

**Calcium mobilization assay.** Calcium mobilization assay to measure GHSR signaling was carried out in HEK-293T cells cultured in DMEM supplemented with 10% (vol/vol) FBS (Fisher Scientific). The cDNAs encoding human GHSR was cloned into the pcDNA3.1(+) vector (Invitrogen) with a FLAG peptide fused to the N terminus. Mutant variants were then generated by site-directed mutagenesis and all mutants were fully sequenced. Various GHSR constructs were transfected to HEK-293T cells using FuGENE Transfection Reagent (Promega). The surface expression of wtGHSR and various mutants in HEK-293T cells was determined and confirmed using FACS with a fluorescent FLAG M1 antibody (homemade).

To measure the agonist-induced calcium release, 50 μL dye loading buffer made of Hank's Balanced Salt Solution (HBSS) with 5 μM Fluo-4 (Sigma), 0.2% pluronicacid, and 1% FBS was added to the GHSR-expressing cells and incubated for 1 h at 37 °C. Cells were then washed twice with HBSS buffer and left at room temperature in 50 μL HBSS. Fluo-4 fluorescence intensity (excitation 480 nm, emission 520 nm) was measured as an indicator of calcium release in a multimode reader (Spark 20 M, TECAN). For the concentration-dependent responses of ghrelin or ibutamoren, 50 μL HBSS buffer containing different concentrations of ligand was injected to each well, and fluorescence was measured constantly in real time for 90 s. The data were analyzed in GraphPad Prism 8 using the one site dose-response stimulation method. Results were presented as mean ± S.D. from three independent experiments.

**GTP turnover assay**. Human GHSR was expressed in *E. coli* inclusion bodies, folded in amphipol A8-35, and then inserted into lipid nanodiscs formed with MSP1E3D1 and POPC:POPG (3:2 molar ratio) containing or not 10% cholesterol, as described in Damian et al.[54]. After insertion into nanodiscs, active receptor fractions were purified using affinity chromatography with the biotinylated version of the JMV2959 antagonist immobilized on a streptavidin column[33]. GHSR containing disks were separated from aggregates and possible trace amounts of the ligand through a size-exclusion chromatography step on a Superdex 200 increase 10/300 GL column (GE Healthcare) using a 25 mM Na-HEPES, 150 mM NaCl, 0.5 mM EDTA, pH 7.5 buffer as the eluent. GTP turnover was assessed as described previously[55]. All experiments were carried out at 15 °C. The receptor was first incubated with the isolated G protein and, when applicable, the ligand (10 μM) for 30 min in a 25 mM HEPES, 100 mM NaCl, 5 mM MgCl₂, pH 7.5 buffer. GTP turnover was then started by adding GTP (5 μM) and GDP (10 μM), and the amount of remaining GTP was assessed after 10 min incubation at 15 °C using the GTPase-Glo assay (Promega). The signal was normalized in each case to that in the absence of receptor (100%).

**Ligand binding assays**. Direct ligand binding experiments were performed using the HTRF signal between GHSR labeled at its N-terminus with Lumi-4 Tb and a ghrelin peptide labeled with dy647 on an additional cysteine positioned at its C-terminus[33]. Titration experiments were carried out with receptor concentrations in the nanomolar range and increasing concentrations in labeled ghrelin. Competition ligand-binding assays were performed using the HTRF signal between GHSR labeled with Lumi-4 Tb and the JMV2959 antagonist labeled with dy647. Increasing concentrations in the competing compound were added to a receptor: antagonist mixture (100 nM concentration range). In all cases, the ligand: receptor mixtures were incubated for 30 min at 15 °C and the HTRF signal recorded using a Cary Eclipse spectrofluorimeter (Varian) with an excitation wavelength set at 337 nm and an emission wavelength at 665 and 620 nm. All binding data were analyzed using Prism 8.0.

**Reporting summary**. Further information on research design is available in the Nature Research Reporting Summary linked to this article.

## Data availability
The 3D cryo-EM density maps of the ghrelin-GHSR-Gᵢ complex and the ibutamoren-GHSR-Gᵢ complex generated in this study have been deposited in the Electron Microscopy Data Bank database under accession codes EMD-24267 and EMD-24268, respectively. The atomic coordinates for the atomic models of the ghrelin-GHSR-Gᵢ and ibutamoren-GHSR-Gᵢ complexes generated in this study have been deposited in the Protein Data Bank database under accession codes 7NA7 and 7NA8, respectively. The raw data for the main Fig. 2c, f and Supplementary Figs. 2b, c, 3b, and 4b generated in this study are provided in the Source Data file. The structural models of GHSR with the antagonist C12 and the neurotensin-NTSR-Gi complex used in this study are available in the Protein Data Bank database under accession codes 6KO5 and 6OS9. Source data are provided with this paper.

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

## Acknowledgements

We thank members of the Cryo-electron Microscopy and Tomography Center of St Jude Children's Research Hospital for help with cryo-EM data collection. We thank grants HL143037 (to J.S.), R35GM128641, and R03TR003306 (to C.Z.) from the National Institutes of Health, USA. We thank funding support from the American Lebanese Syrian Associated Charities (ALSAC) (to J.S.).

## Author contributions

H. L. performed protein expression and purification for cryo-EM studies and the calcium signaling assay. A.M. and J.S. performed cryo-EM data collection and processing. H.L., D.S., and C.Z. built the models and performed structure refinement. M.D. and J.B. performed the GTP turnover and ligand binding assays. H.L., S.J., and C.Z. supervised all studies, and C.Z wrote the manuscript with H.L., J.B., and J.S.

## Competing interests

Dr. Cheng Zhang serves as a consultant for Biogen. The remaining authors declare no competing interests.
