## [Peer Review File · Nature Communications]

Structural basis of human ghrelin receptor signaling by ghrelin and the synthetic agonist ibutamorenREVIEWER COMMENTS

Reviewer #1 (Remarks to the Author):

The authors of the article titled “Structural basis of human ghrelin receptor signaling by ghrelin and the synthetic agonist ibutamoren” report two cryo-electron microscopy structures of human GHSR in complex with Gi and two agonists, ghrelin (peptide, native ligand) and ibutamoren (small molecule). The study reveals a salt bridge (E1243.33 and R2836.55) and an aromatic cluster (W2766.48, F2796.51, H2806.52, and F3127.42) near the agonist-binding pocket as important structural motifs in receptor activation. This is an interesting study and worth of publication in Nature Communications after revising the manuscript according to the following comments:

(1) For the nanodisc GTP turnover assay presented in Extended Data (ED) Figure 2. Does cholesterol change the basal activity of Gi (i.e., non-normalized data for Gi/no ligand should be shown)? Did cholesterol change the ghrelin binding affinity/kinetics?

(2) P4 contradictory sentences: “such a binding mode is different from the predicted ghrelin binding mode based on the previous crystal structure” then “It was suggested that the salt bridge between E1243.33 and R2836.55 (superscripts represent Ballesteros-Weinstein numbering 36) of GHSR divides the ghrelin-binding pocket into two cavities, cavity I and II 34. In our structures, we also observed this salt bridge.” It is hard to follow this paragraph....what is predicted vs what is seen here. A Figure depicting this difference might be helpful.

(3) Fig 2C/F and EDFig3c. Ghrelin/ibutamoren binding assay would be more enlightening to assess these mutants. Ca²⁺ assay might indicate no loss of binding but disrupted conformational change/signal transmission upon binding. Differentiation of these two is important for judging the validity of the authors' hypothesis.

(4) P6 - ECD extension of TM7 – Is there a potential reason why this wasn't seen in the X-ray structures? i.e., crystal packing? Additional mutations (pro-scan) aimed at disrupting this novel helix should be done to determine its functional relevance and whether it is an aberration of the protein prep method the authors performed. From the density map in Fig1 and EDFig5, it is hard to judge the validity of this helical extension; a close-up of this would be useful.

(5) Dynamic nature of Gi-coupling to GHSR – I think the authors are overanalyzing their data here. This whole section should be toned down and not a major conclusion as it currently is (i.e., listed in the abstract as a major finding). It is more an interesting side note that requires further study.

(6) In the prep of the proteins, the authors mentioned that they used 1 μM of PF-05190457 – did the authors note any contamination of PF-05190457 in the final structures?

(7) The authors said they used anti-flag M1 antibody resin (homemade) but gave no details of its production.

Reviewer #2 (Remarks to the Author):

Heng Liu et. al present a manuscript entitled “Structural basis of human ghrelin receptor signaling by ghrelin and the synthetic agonist ibutamoren”. The authors present the cryo-EM structure of the ghrelin receptor in complex with its endogenous agonist ghrelin and the non-peptidic agonist ibutamoren. The authors describe the agonists binding mode, the activation of the ghrelin receptor as well as its coupling to Gi proteins, all supported by mutagenesis and functional assays. The ghrelin receptor has high therapeutic relevance in stress and anxiety and it has been the focus of numerous studies. The structural information generated in this work will undoubtedly have a high impact in future studies. In particular, the ghrelin receptor has the unique properties of binding ghrelin, which is a peptide that needs to be acylated in order to have agonistic activity, and this is an uncommon feature

seen in GPCRs. Both cryo-EM maps are at 2.7 Å resolution, which is better than the average GPCR structure. Additionally the authors were able to isolate a secondary conformation of the Gi heterotrimer when bound to the ghrelin receptor in complex with ghrelin. This has been observed also in the neurotensin receptor and might hint towards information regarding G protein coupling.

I would like to highlight the following points:

1. A second conformation of the Gi coupling was observed for the ghrelin-bound receptor but not for the ibutamoren. Have the authors search for such a conformation in the ibutamoren-bound dataset? If this second conformation is absent, the authors should comment on whether they think this might be ligand specific?
2. Although the resolution of the structures is high for a GPCR, the lower resolution at the ligand binding site (as it is typical for cryo-EM structure of GPCRs) as well as the mobility of some parts of the ligand makes me wonder how confident are the authors when modelling the phenyl group of ibutamoren in cavity II (with the arms with the amine group in cavity II instead). The authors should comment on the facts that lead them to model the ligand in this conformation and state what is their confidence in modelling, or state in the paper that the alternative conformation is possible based on the current experimental map.
3. At the end of the paper it is stated that “Both receptors lack structural elements for selective Gi coupling, which may result in potentially multiple modes of interactions.”. To my knowledge, the elements for selective Gi coupling have not been understood yet (or any other G protein). The authors should clarify what they mean with this sentence.
4. In materials and methods, the protocol for exchanging DDM and LMNG should be more specific. It is only stated that it was exchanged slowly...please provide more detailed information the detergent-exchange protocol.
5. In supplementary figure 1. The local resolution map for the two models show marked differences in resolution while the overall resolution is identical. For example, the ghrelin-bound local resolution map seems to have a local resolution that would yield a higher resolution map than the ibutamoren map. The authors should revise the calculations of the local resolution maps to make sure they are correct.
6. Additionally the authors should provide a FSCwork/FSCfree plot to discard overfitting problems within at the modelling stage.

Manuscript ID: NCOMMS-21-25467-T

Title: **Structural basis of human ghrelin receptor signaling by ghrelin and the synthetic agonist ibutamoren**

We thank all reviewers for their constructive comments. Please see our detailed responses to the comments below. The reviewers' comments are in **blue** font and our responses are in **black** font.

Reviewer #1:

The authors of the article titled “Structural basis of human ghrelin receptor signaling by ghrelin and the synthetic agonist ibutamoren” report two cryo-electron microscopy structures of human GHSR in complex with Gi and two agonists, ghrelin (peptide, native ligand) and ibutamoren (small molecule). The study reveals a salt bridge (E1243.33 and R2836.55) and an aromatic cluster (W2766.48, F2796.51, H2806.52, and F3127.42) near the agonist-binding pocket as important structural motifs in receptor activation. This is an interesting study and worth of publication in Nature Communications after revising the manuscript according to the following comments:

(1) For the nanodisc GTP turnover assay presented in Extended Data (ED) Figure 2. Does cholesterol change the basal activity of Gi (i.e., non-normalized data for Gi/no ligand should be shown)? Did cholesterol change the ghrelin binding affinity/kinetics?

We believe that a figure with unnormalized data would be less conclusive, as there is some variability in the absolute fluorescence values from an experiment to another. We thus had to systematically normalize each replicate to an internal reference, which is the GTP turnover measured with the G protein in the absence of receptor (empty nanodiscs). Our data shown in **Extended Data Figure 2b** clearly suggested that there is a statistically significant effect of cholesterol on basal activity, although this effect is of lower amplitude than that observed in the presence of ghrelin.

We have also provided additional data in **Extended Data Figure 2c** showing that cholesterol could slightly increase the affinity of GHSR for ghrelin (about 2-fold lower K_d value) in the presence of cholesterol.

(2) P4 contradictory sentences: “such a binding mode is different from the predicted ghrelin binding mode based on the previous crystal structure” then “It was suggested that the salt bridge between E1243.33 and R2836.55 (superscripts represent Ballesteros-Weinstein numbering 36) of GHSR divides the ghrelin-binding pocket into two cavities, cavity I and II 34. In our structures, we also observed this salt bridge.” It is hard to follow this paragraph...what is predicted vs what is seen here. A Figure depicting this difference might be helpful.

We thank the reviewer for the suggestion. We did observe a similar salt bridge between E124^{3.33} and R283^{6.55} in both of our structures as that observed in the previously reported structure of antagonist-bound GHSR. However, the binding mode of ghrelin observed in our structure is different from that predicted based on the antagonist-bound GHSR. It was suggested that the octanoyl group of ghrelin samples a hydrophobic crevasse between TM6 and TM7 in the large Cavity I. In our structure, the octanoyl group of ghrelin actually extends into the small Cavity II. In addition, in both of our structures, TM7 shifts towards TM6 compared to that in the inactive GHSR, closing the previously observed hydrophobic crevasse in the antagonist-bound structure.

We have revised this paragraph to make it clearer. However, since we don't have the predicted structural model of GHSR bound to ghrelin, it is difficult to make a figure depicting such difference. We hope that the reviewer will find our revised paragraph clear enough on this point.

(3) Fig 2C/F and EDFig3c. Ghrelin/ibutamoren binding assay would be more enlightening to assess these mutants. Ca²⁺ assay might indicate no loss of binding but disrupted conformational change/signal transmission upon binding. Differentiation of these two is important for judging the validity of the authors' hypothesis.

We thank the reviewer for the suggestion. We have now included ghrelin binding data in **Supplementary Figure 3b**. Our data showed that individual mutations of hydrophobic residues that interact with the octanoyl moiety of ghrelin to alanine led to decreased affinity of GHSR for ghrelin. Also, mutations of the salt bridge residues E124^{3,33} and R283^{6,55} to either alanine or glutamine could almost abolish ghrelin binding. We have included discussion of the binding data in the third paragraph of section 'Ghrelin and ibutamoren recognition' in our revised manuscript.

(4) P6 - ECD extension of TM7 – Is there a potential reason why this wasn't seen in the X-ray structures? i.e., crystal packing? Additional mutations (pro-scan) aimed at disrupting this novel helix should be done to determine its functional relevance and whether it is an aberration of the protein prep method the authors performed. From the density map in Fig1 and EDFig5, it is hard to judge the validity of this helical extension; a close-up of this would be useful.

We thank the reviewer for the suggestion. We have included a figure in **Extended Data Figure 4a** showing the density map of the extracellular part of TM7. We also performed Pro-scan of four residues at the extracellular region of TM7 and measured ghrelin-induced GHSR signaling on the mutants. The results showed that individual mutations at this region didn't significantly affect the agonistic action of ghrelin, suggesting little involvement of this region in ghrelin binding. We have included the mutagenesis data **Extended Data Figure 4b** and discussed the data in the first paragraph of section 'Distinctive activation mechanism of GHSR' in our revised manuscript. The disordered extracellular region of TM7 in the crystal structure of inactive GHSR might be caused by crystal packing (**Figure R1**). We now stated that 'it is possible that the disordered extracellular region of TM7 observed in the crystal structure of inactive GHSR was caused by crystal packing' in our revised manuscript.

Figure R1. The disordered extracellular region of TM7 (circled) of inactive GHSR is at the interface of one GHSR molecule (magenta) and a neighboring symmetric molecule (green). The narrow space may not allow extended helical structure of TM7.

(5) Dynamic nature of G_i -coupling to GHSR – I think the authors are overanalyzing their data here. This whole section should be toned down and not a major conclusion as it currently is (i.e., listed in the abstract as a major finding). It is more an interesting side note that requires further study.

We thank the reviewer for the suggestion. Our cryo-EM data analysis revealed at least two major structural classes for the ghrelin-GHSR- G_i particles, and we obtained high-resolution maps for both structures. The results unambiguously showed two distinct binding modes of G_i to GHSR. We found it interesting since to the best of our knowledge similar results have been reported only for NTSR1 but no other G_i -coupled GPCRs before.

We have toned down our discussion of this observation in the revised manuscript as suggested by the reviewer. We removed the sentence "indicating highly dynamic nature of G_i -coupling to GHSR" in the abstract. We also changed the title "Dynamic nature of G_i -coupling to GHSR" to "Molecular details of G_i -coupling to GHSR" and stated that the structural variations we observed might be caused by the non-native buffer conditions of our samples in the cryo-EM studies.

(6) In the prep of the proteins, the authors mentioned that they used 1 μ M of PF-05190457 – did the authors note any contamination of PF-05190457 in the final structures?

We used the GHSR inverse agonist PF-05190457 during protein expression and early stages of protein purification to stabilize GHSR. After loading purified GHSR to an anti-FLAG M1 antibody column, we washed the resin extensively with buffers containing agonists to exchange the ligand from PF-05190457 to agonists bound in GHSR. We also included agonists in all buffers after this step. We didn't observe any density map in our structural analysis suggesting PF-05190457 contamination. Also, GHSR bound to PF-05190457 is not likely to form a stable complex with G_i .

(7) The authors said they used anti-flag M1 antibody resin (homemade) but gave no details of its production.

To make the resin, we immobilized purified M1 antibody onto a CNBr-activated resin purchased from Cytiva Life Sciences. Antibodies can be coupled to the resin through covalent linking of free $-NH_2$ groups to the resin matrix. We have included such information in our revised manuscript.

Reviewer #2 (Remarks to the Author):

Heng Liu et. al present a manuscript entitled “Structural basis of human ghrelin receptor signaling by ghrelin and the synthetic agonist ibutamoren”. The authors present the cryo-EM structure of the ghrelin receptor in complex with its endogenous agonist ghrelin and the non-peptidic agonist ibutamoren. The authors describe the agonists binding mode, the activation of the ghrelin receptor as well as its coupling to G_i proteins, all supported by mutagenesis and functional assays. The ghrelin receptor has high therapeutic relevance in stress and anxiety and it has been the focus of numerous studies. The structural information generated in this work will undoubtedly have a high impact in future studies. In particular, the ghrelin receptor has the unique properties of binding ghrelin, which is a peptide that needs to be acylated in order to have agonistic activity, and this is an uncommon feature seen in GPCRs. Both cryo-EM maps are at 2.7 Å resolution, which is better than the

average GPCR structure. Additionally the author were able to isolate a secondary conformation of the Gi heterotrimer when bound to the ghrelin receptor in complex with ghrelin. This has been observed also in the neurotensin receptor and might hint towards information regarding G protein coupling.

I would like to highlight the following points:

1. A second conformation of the Gi coupling was observed for the ghrelin-bound receptor but not for the ibutamoren. Have the authors search for such a conformation in the ibutamoren-bound dataset? If this second conformation is absent, the authors should comment on whether they think this might be ligand specific?

We thank the reviewer for the comment. Actually, we did observe a second conformation for the ibutamoren-GHSR-G_i complex. Please see **Extended Data Figure 1d**. We have also included the information in our revised manuscript.

2. Although the resolution of the structures is high for a GPCR, the lower resolution at the ligand binding site (as it is typical for cryo-EM structure of GPCRs) as well as the mobility of some parts of the ligand makes me wonder how confident are the authors when modelling the phenyl group of ibutamoren in cavity II (with the arms with the amine group in cavity II instead). The authors should comment on the facts that lead them to model the ligand in this conformation and state what is their confident in modelling, or state in the paper that the alternative conformation is possible based on the current experimental map.

We thank the reviewer for the suggestion. We have tried many different conformations of ibutamoren during structure refinement. The alternative conformation of ibutamoren with the small arm with an amine group occupying cavity II would place the phenyl group at the bottom of cavity I, which doesn't have enough space to accommodate the phenyl group. As shown in **Figure R2b** below, this alternative conformation of ibutamoren would cause severe clash with S123. Also, the polar environment of the bottom region of cavity I is not suitable for the phenyl

Figure R2. (a) Conformation of ibutamoren depicted in our structure. (b) Alternative conformation of ibutamoren with the smallest arm occupying cavity II.

group. We believe that the current conformation of ibutamoren depicted in our structure is the most likely one based on the density map. Nevertheless, in our revised manuscript, we stated that subtle structural variations of ibutamoren are possible due to the limitation of resolution of the density map at the end of the first paragraph of section 'Structure determination of two GHSR-G_i complexes and overall structures'.

3. At the end of the paper it is stated that “Both receptors lack structural elements for selective G_i coupling, which may result in potentially multiple modes of interactions.”. To my knowledge, the elements for selective G_i coupling have not been understood yet (or any other G protein). The authors should clarify what they mean with this sentence.

We meant that since both GHSR and NTSR can signal through other G proteins besides G_i, they do not have a high selectivity for G_i and thus they do not have any potential structural elements leading to selective G_i coupling. Nevertheless, as suggested by Reviewer 1, we have removed this sentence in our revised manuscript.

4. In materials and methods, the protocol for exchanging DDM and LMNG should be more specific. It is only stated that it was exchanged slowly...please provide more detailed information the detergent-exchange protocol.

We have included detailed information of buffer exchange in the revised Methods section.

5. In supplementary figure 1. The local resolution map for the two models show marked differences in resolution while the overall resolution is identical. For example, the ghrelin-bound local resolution map seems to have a local resolution that would yield a higher resolution map than the ibutamoren map. The authors should revise the calculations of the local resolution maps to make sure they are correct.

We thank the reviewer for the suggestion. We used the localres program to calculate the local resolution maps for the ghrelin-GHSR-G_i complex, which may have been overly optimistic in local-resolution estimation. We now switched to a different program, blocres, to calculate the local resolution maps. The results showed similar local-resolution estimations for the ghrelin- and ibutamoren-bound GHSR-G_i complexes as expected. We have revised the **Extended Data Figure 1** accordingly.

6. Additionally the authors should provide a FSCwork/FSCfree plot to discard overfitting problems within at the modelling stage.

As suggested by the reviewer, we have included FSC curves for cross-validation between structural models and cryo-EM maps in **Extended Data Figure 1g and h**. FSC curves of structural models against the half maps and the full map showed very subtle differences, indicating minimal overfitting problems in our structural modeling and refinement. Please also see our description in the figure legends of **Extended Data Figure 1g and h**.

REVIEWER COMMENTS

Reviewer #2 (Remarks to the Author):

The authors have satisfied all the concerns.